# Off-policy Evaluation for Multiple Actions in the Presence of Unobserved Confounders

## Abstract

Off-policy evaluation (OPE) is a crucial problem in reinforcement learning (RL), where the goal is to estimate the long-term cumulative reward of a target policy using historical data generated by a potentially different behaviour policy. In many real-world applications, such as precision medicine and recommendation systems, unobserved confounders may influence the action, reward, and state transition dynamics, which leads to biased estimates if not properly addressed. While existing methods for handling unobserved confounders in OPE focus on single-action settings, they are less effective in multi-action scenarios commonly found in practical applications, where an agent can take multiple actions simultaneously. In this paper, we propose a novel auxiliary variable-aided method for OPE in multi-action settings with unobserved confounders. Our approach overcomes the limitations of traditional auxiliary variable methods for multi-action scenarios by requiring only a single auxiliary variable, relaxing the need for as many auxiliary variables as the actions. Through theoretical analysis, we prove that our method provides an unbiased estimation of the target policy value. Empirical evaluations demonstrate that our estimator achieves better performance compared to existing baseline methods, highlighting its effectiveness and reliability in addressing unobserved confounders in multi-action OPE settings.

## CCS Concepts

• **Information systems → Evaluation of retrieval results**.

## Keywords

Off-policy Evaluation, Multiple Actions, Unobserved confounder, Treatment Recommendation, Deconfounding

**ACM Reference Format:**
Anonymous Author(s). 2018. Off-policy Evaluation for Multiple Actions in the Presence of Unobserved Confounders. In *Proceedings of Make sure to enter the correct conference title from your rights confirmation emai (Conference acronym 'XX)*. ACM, New York, NY, USA, 12 pages. https://doi.org/XXXXXXX.XXXXXXX

## 1 Introduction

Off-policy evaluation (OPE) is important for decision-making under uncertainty, and it is a key topic in reinforcement learning (RL). Unlike online RL, where policies are evaluated in real-time, in OPE

we aim to estimate the long-term cumulative reward of a new policy (i.e., *target policy*) by using historical data generated by a potentially different policy (i.e., *behaviour policy*). The ability to evaluate the performance of a new policy without implementing it is critical, particularly in scenarios where real-world experimentation is costly, risky, or unethical. Such scenarios include precision medicine [27], robotics [53], and recommendation systems [9].

In general, most OPE methods assume the absence of unobserved confounders that affect both actions and rewards, or both actions and next states. This assumption is typically referred to as *Unconfoundedness* [4, 62]. However, in certain real-world applications, this assumption may not hold and unobserved confounders can introduce bias in OPE. For instance, in Figure 1, we illustrate the task of evaluating a new treatment regimen for a patient with diabetes before prescribing it to patients. Doctors would like to first assess the treatment regimen using past clinical records, which include patient health status, prescribed anti-diabetic medications, and blood glucose levels, rather than directly offering uncertain advice that could potentially harm patients. The value of interest is the long-run average deviation from ideal glucose levels. However, there may be unrecorded events, such as the patient's food intake and exercise, that could simultaneously influence the patient's medication routine, blood glucose levels, and future health status. Such unrecorded factors introduce bias and violate the Unconfoundedness assumption. Most existing OPE methods fail to account for such unobserved confounders (see Section 2.1 for details). If these methods are applied to evaluate the new treatment regimen without considering unobserved confounders, it may result in biased evaluations, which may lead to harmful treatment decisions.

Recently, there have been several attempts to apply causal inference to OPE to address the issue of unobserved confounders (see Section 2.2 for details). However, these efforts typically focus on OPE in a single-action setting (i.e., only one single action is taken by an agent at each time step). They do not account for the multi-action setting, which is commonly encountered in the OPE literature [7, 69]. Consider the diabetes treatment example: patients are often prescribed multiple anti-diabetic drugs simultaneously, such as Sulfonylureas, Biguanides, and/or DPP-4 inhibitors [63]. The schedule of medication intake may be influenced by the unmeasured diet and exercise of a patient. Furthermore, these existing methods for the single-action setting often rely on auxiliary variables (e.g., instruments [35, 67], confounder proxies [4, 5]) to achieve an unbiased evaluation of the target policy in the presence of unobserved confounders. However, if one intends to simply extend these conventional auxiliary variables-based methods to the multi-action setup, it introduces additional challenges. For instance, the instrumental variable approach requires at least as many instrumental variables as the number of actions [45]; the confounder proxy approach requires outcome-inducing proxies to be causally uncorrelated with all actions [60]. In practical applications, identifying valid auxiliary variables that meet the above requirements is

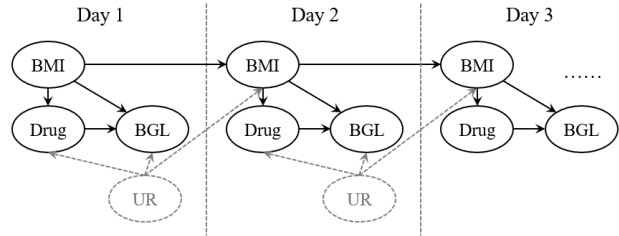

**Figure 1: A graphical representation of a proposed treatment regimen for diabetes patients, where *BMI* represents a patient's body mass index (health status), *Drug* represents the drug administered, and *BGL* represents the patient's blood glucose level. In this treatment plan, *UR* represents the unrecorded food intake or exercise of the patient that affects medication routine, fluctuations in blood glucose level, and the patient's body mass index the next day.**

quite difficult. Therefore, it is necessary to develop a method that not only fills the aforementioned gap in the multi-action setting but also imposes less restrictive requirements on auxiliary variables.

In the causal inference area, a few studies have attempted to address the problem of multi-treatment unmeasured confounding currently [16, 26, 66]. These efforts, however, suffer from either the absence of theoretical guarantees for identification or the requirement of an infinite number of treatments to mitigate confounding. We note that Miao et al. [41] recently developed a novel auxiliary variable method to address the bias introduced by multi-treatment unmeasured confounding. This work establishes strict theoretical guarantees for identification under a more general, though not unrestricted treatment-confounder model, thus avoiding the aforementioned drawbacks. Motivated by this work, we propose an auxiliary variable-aided method for OPE in the multi-action setting with unmeasured confounding. Unlike the traditional auxiliary variable methods, our method does not require as many auxiliary variables as there are actions, nor does it necessitate confounder proxies that are unrelated to all actions; instead, a single auxiliary variable is sufficient to complete the identification of policy value. This relaxation of the traditional assumption makes the approach more feasible in practical applications. While our method is inspired by [41], it is important to note that adjusting for unobserved confounders in the OPE setting is more complex than in the static setting, as the confounder in the OPE setting may influence both the immediate rewards and next states at each step, and the state transition involved in OPE amplifies the difficulty of adjusting the accumulated bias. Our contribution can be summarised as the following:

- We propose an auxiliary variables based method for OPE in a multi-action setting, which can achieve an unbiased estimation of the target policy in the presence of unobserved confounders. To the best of our knowledge, this is one of the first works on off-policy evaluation in the multi-action setup with unobserved confounders.
- Through comprehensive theoretical analysis, we demonstrate that the proposed method provides an unbiased estimation

of the target policy value in the presence of unobserved confounders. Additionally, we develop a direct value estimator as part of the proposed method.
- In simulation experiments and a treatment recommendation example driven by a real OPE application, the proposed method achieves better empirical performance compared to other baseline methods, indicating its effectiveness and reliability. The source code can be found at https://anonymous.4open.science/r/multi-action-OPE-with-UC-E661.

## 2 Related Work

### 2.1 Off-policy Evaluation

Over the past several years, OPE has been extensively studied in reinforcement learning (RL). Current OPE methods can be broadly categorised into four types [33, 52, 62]. The first category is the **importance sampling** (IS) based methods [11, 18–20, 56, 61]. This type of method computes the ratio of the probabilities of trajectories under the target policy to those under the behaviour policy and use these ratios to adjust the observed rewards. One advantage of IS methods is that they do not require modelling the dynamics of the environment or estimating the reward function. However, they are prone to high variance, especially in long-horizon tasks where the product of probabilities can become very small or unstable. The second category is the **direct methods** (DMs) [15, 31, 32, 36, 38, 57]. This type of methods directly estimate the Q-function or the value function of a target policy from the historical data. DMs have lower variance compared to IS, but their performance heavily depends on the accuracy of the model used to estimate the Q-function or value function. The third category is the **doubly robust** (DR) methods [14, 22, 59], which combines the strengths of both IS and DMs to achieve a balance between bias and variance. This type of method uses IS to account for discrepancies between the behaviour and target policies, while relying on the DMs for value estimation. It retains unbiasedness even when either the importance weights or the value function estimates are imprecise, provided that at least one of them is accurately estimated. Distinct from these three model-free methods, the fourth category is **model-based** methods [12, 29, 65], which focus on explicitly constructing a model of the environment's dynamics, such as the state transition function and reward function, to estimate the value of a target policy. By modelling the underlying environment, this type of method simulates trajectories under the target policy, allowing for the evaluation of policy performance.

All of these methods implicitly rely on the Unconfoundedness assumption, yet unobserved confounders are often unavoidable in the real-world. Recently, several methods have been developed to address this issue. In the next section, we review existing works that focus on handling unobserved confounders in OPE.

### 2.2 OPE with unobserved confounders

To address the issue of unobserved confounders, researchers have increasingly incorporated causal inference techniques with existing OPE methods to achieve unbiased policy evaluation. These methods can be broadly categorised into two directions. The first direction employs sensitivity analysis or relies on weak assumptions to develop identification bounds on the value of the target policy [6, 23, 25, 44, 68]. However, these methods depend on specific

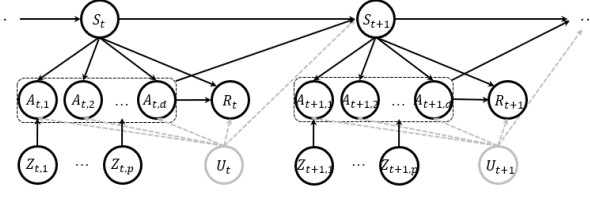

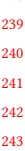

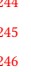

**Figure 2: Causal diagram for MMDPUC, where $U_t$ denotes the unobserved confounders that may affect both $A_t$ and $R_t$ / $S_{t+1}$.**

assumptions that may vary across different settings, potentially leading to inconsistent results. The second direction utilises auxiliary variables from the observed data to adjust for bias introduced by unobserved confounders, yielding unbiased estimates of the target policy. The most commonly used auxiliary variables include instrumental variables [35, 67], confounder proxies [4, 5]. The core principle of these methods is to mitigate the confounding interference in the causal relationship between treatment and outcome by introducing variables that are either independent of the confounder but related to the outcome or that capture information about the confounder. Consequently, the quality of the chosen auxiliary variables plays a critical role in determining the accuracy of the evaluation.

However, the aforementioned approaches share a common limitation: they focus exclusively on OPE in a single-action setting with unobserved confounders, where the behaviour or target policy takes only one action at each time step. They do not address scenarios where multiple actions are confounded. In addition, if one were to simplify a multi-action problem as a multiple single-action problem by treating it as a vector-valued action, applying the method above directly to a multi-action scenario would impose some additional requirements. For instance, in the instrumental variable approach, the *completeness* assumption requires [45] that the number of instrumental variables must match the number of actions, with each instrument corresponding to a specific action. In the case of confounder proxy methods, the outcome-inducing proxies are required to be causally uncorrelated with all actions [60]. These additional assumptions restrict the applicability of these methods in real-world settings.

Recently, there have been a few attempts to identify the effects of multiple treatments on outcomes in the presence of unobserved confounders. Wang and Blei [66] were among the first to provide an intuitive justification for addressing multi-treatment unmeasured confounding. They utilized a factor model to estimate the confounder among multiple treatments and employed the confounder estimate to adjust for bias. However, as discussed in these works [21, 47], the effect of multiple treatments in this work cannot be uniquely determined from observed data. Kong et al. [26] utilized a linear factor model to address the confounding among multiple treatments, but this approach is only applicable to binary outcomes and cannot be generalized to more complex outcome models. Grimmer et al. [16] considered a linear outcome model with multiple treatments that are confounded or mismeasured. However, this

approach requires an infinite number of treatments to guarantee identification of the results. We note that Miao et al. [41] model multiple treatments using factor models and achieve unique identification of multiple treatment effects by leveraging a small number of auxiliary variables. Unlike the aforementioned methods, this approach not only provides rigorous theoretical guarantees for identification but also avoids limitations such as binary outcome models and the requirement of infinite treatment numbers. Our paper extends this theory to the context of multi-action policy evaluation and provides unbiased estimation of the target policy's value.

## 3 Preliminaries

### 3.1 Data-Generating Process

We consider the observational data generated from a Markov Decision Process with multiple actions in the presence of unobserved confounders (MMDPUC), which is a confounded generalization of the Multi-action Markov Decision Process (MMDP) [64], as illustrated in Figure 2. A single trajectory under the MMDPUC is represented by a sequence of tuples $(S_t, A_t, U_t, R_t)$ at the $t$-th step for any $t \in \mathcal{T}$. Here, $S_t \in \mathcal{S}$ denotes the state, $A_t = (A_{t,1}, \ldots, A_{t,d}) \in \mathcal{A} = (\mathcal{A}_1 \times \ldots \times \mathcal{A}_d)$ is a vector of $d$ actions, and $R_t \in \mathcal{R}$ represents the immediate reward. The calligraphic letters represent the value spaces of the corresponding variables. Let $U_t$ denote the set of unobserved confounders at step $t$, which confounds the relationship between $A_t$ and $R_t$, as well as $A_t$ and $S_{t+1}$. For example, in the context of diabetes treatment, $S_t$ corresponds to a patient's body mass index at step $t$, $A_t$ refers to the administration of multiple anti-diabetic drugs, $R_t$ represents blood glucose levels, and $U_t$ accounts for unmeasured factors such as drug resistance of the patient.

We assume that each historical trajectory follows a common behaviour policy, $\pi_b$, which depends on unobserved confounders $U_t$. The policy $\pi_b(A_{t,1}, \ldots, A_{t,d}|S_t, U_t)$ represents the probability of the agent taking multiple actions given the state $S_t$ and confounders $U_t$, i.e., $\pi_b(a_{t,1}, \ldots, a_{t,d}|s_t, u_t) = p_a(A_{t,1}, \ldots, A_{t,d} = a_{t,1}, \ldots, a_{t,d}|S_t = s_t, U_t = u_t)$. The agent then receives a reward, $R_t \sim p_r(\cdot|S_t, A_t, U_t)$, and transitions to the next state, $S_{t+1} \sim p_s(\cdot|S_t, A_t, U_t)$.

We also need to make some common assumptions as in Off-Policy Evaluation (OPE) literature [43, 44, 54] for the above data-generating process. Let $a_{1:t} = (a_1, a_2, \cdots, a_t)$ denote a sequence of historical actions from time 1 to time $t$. For any sequence of actions $a_{1:t}$, let $R_t(a_{1:t})$ and $S_{t+1}(a_{1:t})$ denote the *potential* immediate reward and the *potential* next state, respectively, that would be observed at time step $t$ if the agent had taken the sequence of actions $a_{1:t}$ up to that point. Let $H_t$ represent the set of all possible histories, defined as $H_t := H_t(a_{1:t-1}) = (S_1, A_1 = a_1, R_1(a_1), S_2(a_1), \cdots, A_{t-1} = a_{t-1}, R_{t-1}(a_{1:t-1}), S_t(a_{1:t-1}))$.

Based on this, we assume:

ASSUMPTION 1. *For any step $t \in \{1, \ldots, T\}$,*

*i Consistency: When $A_{1:t} = a_{1:t}$, we have $R_t = R_t(a_{1:t})$ and $S_{t+1} = S_{t+1}(a_{1:t})$.*

*ii Sequential Ignorability: $R_t(a_{1:t}) \perp A_t|(H_t, U_t)$ and $S_{t+1}(a_{1:t}) \perp A_t|(H_t, U_t)$ for all $a_{1:t}$.*

*iii Positivity (Overlap): $0 < p(A_{t,1} = a_{t,1}, \ldots, A_{t,d} = a_{t,d}|H_t, U_t = u_t) < 1$ for all $a_{t,1}, \ldots, a_{t,d}, u_t$.*

These are the common assumptions for ensuring unbiased OPE in the presence of unobserved confounders [34, 46, 51]. Essentially, the assumptions are a natural extension of the counterfactual or potential outcomes framework [55] widely used in causal inference to decision-making processes based on MDP. *Consistency* states that the observed reward and next state are realizations of the potential outcomes under the actions that were actually taken. *Sequential ignorability* posits that the actions taken are independent of the potential outcomes, $R_t(a_{1:t})$ and $S_{t+1}(a_{1:t})$, conditional on the historical data generated by the policy. This ensures that action assignments are effectively randomized given the history and implies that $U_t$ suffices to control for all confounding at any time step $t$. *Positivity*, also known as "overlap," asserts that all actions assigned have a positive probability of occurring given any history.

## 3.2 Problem Formulation

The complete data consist of N i.i.d. trajectories, given by

$$D = \{(S_{i,t}, A_{i,t}, U_{i,t}, R_{i,t}, S_{i,t+1})\}_{t=1}^{T}, i = \{1, \cdots, N\}, \quad (1)$$

where $T$ denotes termination time step for a single trajectory and all trajectories have the same time step horizon. Note that the dataset $D$ is obtained by following the behaviour policy $\pi_b$.

Let $\pi_t$ denote a deterministic policy to be evaluated (target policy). Under $\pi_t$, at each time $t$, the agent will set $A_t = (a_{t,1}, \ldots, a_{t,d})$ with probability $\pi_t(a_{t,1}, \ldots, a_{t,d}|S_t)$. We define the value function $V^{\pi_t}(S_0)$ for the target policy as the expected cumulative reward over a finite time horizon $T$, obtained by following the target policy $\pi_t$ starting from the initial state $S_0$:

$$V^{\pi_t}(s_0) = \frac{1}{T} \sum_{t=0}^{T} \mathbb{E}^{\pi_t}[R_t|S_0 = s_0], \quad (2)$$

where $\mathbb{E}^{\pi_t}$ denotes the expectation of potential outcome of the immediate reward $R_t$ under $\pi_t$ at time step $t$.

Based on the data that can be observed in Equation 1, our objective is to evaluate the aggregated value:

$$\eta^{\pi_t} = \mathbb{E}_{S_0 \sim \nu}[V^{\pi_t}(S_0)], \quad (3)$$

where the expectation is taken with respect to $\nu$, the distribution over the initial state $S_0$.

We note that we adopt an average reward formulation to address the policy evaluation problem, which is well-suited to specific applications such as precision medicine or treatment recommendation. Additionally, our approach can be easily extended to the discounted reward setting (see A.2 in the Appendix).

## 3.3 Challenges of evaluating policy value

In this section, we discuss the challenges in evaluating the value of the target policy in Equation 2 in the presence of unmeasured confounders.

To begin with, we introduce the do-operator, *do*, to represent an intervention [49]. In causal inference, $do(X = x)$ denotes one exogenously intervenes on the variable $X$, setting it explicitly to $x$, rather than observing its natural occurrence at $x$, without altering the causal relationships among other variables in the system. In the context of OPE, this corresponds to setting the action value to $(a_{t,1}, \ldots, a_{t,d})$ following the target policy while keeping

other functional mechanisms unchanged. For instance, the notation $do(A_t = \pi_t(s_t))$ means that the actions $A_t = (A_{t,1}, \ldots, A_{t,d})$ are set to the value $\pi_t(s_t)$, where $\pi_t(s_t)$ denote the actions that agent takes after observing the state $s_t$ according to the policy $\pi_t$. Note that unlike the behavior policy $\pi_b$, the target policy $\pi_t$ does not depend on the unmeasured confounders $U_t$. This is because the do-operator removes all edges pointing to the intervention node, except for the edge from $S_t$ to $A_t$. In other words, any relationship between $U_t$ and $A_t$ during the data generation process is no longer in effect once we perform the intervention.

Using the do-operator, the expectation of the immediate reward at step $t$ can be expressed according to Markov property as:

$$\mathbb{E}^{\pi_t}[R_t|S_0 = s_0]$$
$$= \mathbb{E}[R_t|do(A_{j,1} = \pi_t(s_j), \ldots, A_{j,d} = \pi_t(s_j)), \forall 0 \leq j \leq t, S_0 = s_0]$$
$$= \mathbb{E}[\mathbb{E}\{R_t|do(A_{t,1} = \pi_t(s_t), \ldots, A_{t,d} = \pi_t(s_t)), S_t, U_t\}|$$
$$do(A_{j,1} = \pi_t(s_j), \ldots, A_{j,d} = \pi_t(s_j)), \forall 0 \leq j < t, S_0 = s_0].$$

If we were able to observe the confounder $U_t$, Assumption 1 would allow for the identification of $\mathbb{E}\{R_t|do(A_{t,1} = \pi_t(s_t), \ldots, A_{t,d} = \pi_t(s_t)), S_t, U_t\}$ using the back-door adjustment [48]:

$$\mathbb{E}\{R_t|do(A_{t,1} = \pi_t(s_t), \ldots, A_{t,d} = \pi_t(s_t)), S_t, U_t\}$$
$$= \sum_{r_t, s_t, a_{t,1}, \ldots, a_{t,d}, u_t} r_t p_r(r_t|s_t, a_{t,1}, \ldots, a_{t,d}, u_t) p_s(s_t) p_u(u_t). \quad (4)$$

However, when $U_t$ is not observed, all the information contained in the observed data is captured by $p(s_t, a_{t,1}, \ldots, a_{t,d}, r_t)$, from which one cannot uniquely determine $\mathbb{E}\{R_t|do(A_{t,1} = \pi_t(s_t), \ldots, A_{t,d} = \pi_t(s_t)), S_t, U_t\}$. Similarly, the reward at the previous time step $j$, $\forall 0 \leq j < t$, cannot also be uniquely determined. Furthermore, as shown in the causal graph in Figure 2, $S_{t+1}$ and $R_t$ share the same causal hierarchy, leading to a similar identification problem for the next state. Therefore, direct application of conventional OPE methods, as discussed in Section 2.1, will result in a biased evaluation of the target policy value in the presence of unmeasured confounders.

# 4 Identification of policy value

In this section, we address the unbiased estimation of the target policy value in the multi-action setup in the presence of unobserved confounders using auxiliary variables. First, we introduce the relevant assumptions regarding these auxiliary variables. Next, we derive a formulaic expression for the value function $V^{\pi_t}(s_0)$ with the aid of auxiliary variables, which allows for unbiased estimation of the target policy value given observed data even in the presence of unobserved confounders. This result can serve as a basis for the value estimator we proposed in Section 5.

## 4.1 The Auxiliary Variables Assumption

From Equation 4, we can infer that the lack of identification of potential immediate reward is due to the unknown distribution $p_r(r_t|s_t, a_{t,1}, \ldots, a_{t,d}, u_t)$ and $p_u(u_t)$ distribution. For any possible distributions $p_r$ and $p_u$ without imposing additional assumptions, this would result in different potential rewards. Therefore, we introduce auxiliary variables and impose extra assumptions to achieve the unique identification of the above distribution.

We assume that there exists a vector of observed auxiliary variables at each time step $t$, which may consist of a single variable, denoted by $Z_t$ in Figure 2. The observed data distribution at each time step $t$ is captured by $p(s_t, a_t, r_t, z_t)$, from which we aim to identify the potential reward distribution $p_r(r_t|s_t, a_t, u_t)$ and the state transition distribution $p_s(s_{t+1}|s_t, a_t, u_t)$. Given $z_t$, we let $p_{s,a,u}(s_t, a_t, u_t|z_t)$ represent the state-action-confounder distribution, and $p_{s,a}(s_t, a_t|z_t)$ the marginalized distribution over $u_t$. Let $p_u(u_t|s_t, a_t, z_t)$ denote the confounder distribution conditional on $s_t$, $a_t$, and $z_t$. The auxiliary variables $Z_t$ rest on the following assumption:

ASSUMPTION 2. *For any step $t \in \{1, \ldots, T\}$,*

  *i Exclusion restriction:$Z_t \perp R_t|(A_t, S_t, U_t), Z_t \perp S_{t+1}|(A_t, S_t, U_t)$.*

  *ii Equivalence: For any $p_{s,a,u}(s_t, a_t, u_t|z_t)$ that solves $p_{s,a}(s_t, a_t|z_t) = \sum_{u_t} p_{s,a,u}(s_t, a_t, u_t|z_t)$ can be written as $p_{s,a,u}(s_t, a_t, u_t|z_t) = p(S_t = s_t, A_t = a_t, g(U_t) = u_t|z_t)$, where $g$ denotes any function that is invertible but not necessarily to known.*

  *iii Completeness: For any $p_u(u_t|s_t, a_t, z_t)$, $p_u(u_t|s_t, a_t, z_t)$ is complete in $z_t$, that is, for any fixed $s_t$ and $a_t$, $E(h(U_t)|S_t = s_t, A_t = a_t, Z_t) = 0, \forall Z_t$ almost surely if and only if $h(U_t) = 0$ almost surely, where $h$ is any family of function in $L^2$.*

The exclusion restriction implies that the auxiliary variables $Z_t$ should only affect the reward $R_t$ and the next state $S_{t+1}$ indirectly through the actions $A_t$. This assumption rules out the existence of directed edges from $Z_t$ to $R_t$ and $S_{t+1}$ in Figure 2. It is analogous to the exclusion restriction assumption in instrumental variables [3] and treatment-inducing confounder proxies [40, 60].

Equivalence implies that the state-action-confounder distribution is based on a model identified by a one-to-one transformation of $U_t$, which restricts the class of state-action-confounder distributions. Specifically, this assumption requires that the dimension of the confounder $U_t$ be smaller than that of the actions $A_t$. The purpose of this restriction is to enable the use of factor models or mixture models to describe the relationships between $S_t$, $A_t$, and $U_t$. Identification results for factor or mixture models have been widely applied in causal effect estimation [1, 30, 39, 66].

Completeness is a fundamental concept in causal inference and statistical inference, primitive conditions are readily available in some literature [2, 10, 45]. Here, it can be interpreted as the notion that most of the information or randomness in the unmeasured confounders $U_t$ is captured by the variables $(S_t, A_t, Z_t)$. Specifically, the completeness assumption means that conditional on $S_t$ and $A_t$, any variability in $U_t$ is reflected in the variability of $Z_t$, which is analogous to the relevance condition in instrumental variable identification. This concept is easiest to understand when both $U_t$ and $Z_t$ are categorical, with dimensions $d_u$ and $d_z$, respectively. In this case, completeness requires that $d_z \geq d_u$. In practice, completeness is more plausible when practitioners measure a rich set of potential auxiliary variables for confounding adjustment. Typically, when the dimension of $U_t$ is much smaller than that of $A_t$, the dimension of $Z_t$ can also remain small.

## 4.2 Identification of policy value

In this section, we demonstrate that $V^{\pi_t}(s_0)$ can be estimated unbiasedly from the observed data even in the presence of unobserved confounders, as shown in Theorem 4.1 below.

THEOREM 4.1. *Under Assumptions 1 - 2, $V^{\pi_t}(s_0)$ equals*

$$\frac{1}{T}\sum_{t=0}^{T}\sum_{\tau_t} r_t\{\prod_{j=0}^{t} p_{s,r}(s_{j+1}, r_j|s_j, a_j, u_j)p_{s,a,u}(s_j, a_j, u_j|z_j)p_z(z_j)\},$$

*where $\tau_t$ denote the historical data $\{(s_j, z_j, a_j, r_j)\}_{j=0}^{t}$ up to time $t$.*

PROOF. See Appendix A.1. □

*Remark 1.* The main idea of the proof of Theorem 4.1 relies on first applying the Markov property to decompose the identification problem of the long-term cumulative reward into a sequence of single-stage problems. Then, we iteratively apply the potential outcomes framework (Assumption 1) and the conditions related to auxiliary variables (Assumption 2) to estimate the potential reward under the target policy using the observed data.

*Remark 2.* The key aspect of the identification process is to uniquely determine the state transition distribution $p_s(s_{j+1}|s_j, a_j, u_j)$ and the reward distribution $p_r(r_j|s_j, a_j, u_j)$ from the observed data. By leveraging auxiliary variables that satisfy Assumption 2, we can achieve unique identification of these distributions. It is important to note that, unlike the back-door adjustment, we do not identify the true state transition and reward distributions but instead obtain arbitrary distributions that satisfy Assumption 2 (iii).

*Remark 3.* Theorem 4.1 outlines three steps in the auxiliary variable approach at each time step. First, we estimate the distribution of each observed variable by using standard density estimation techniques and the confounder distribution $p_u(u_j|s_j, a_j, z_j)$ by using a standard factor model. Note that we do not identify the true distribution $p_u$, but some invertible transformation $g(U_t)$. Next, we identify the state transition distribution $p_s(s_{j+1}|s_j, a_j, u_j)$ and the reward distribution $p_r(r_j|s_j, a_j, u_j)$ by solving Equation 12 in the Appendix. Finally, we integrate the distributions obtained in the first two steps to estimate $\mathbb{E}\{R_j|S_j, do(A_j = \pi_t(S_j)), U_j\}$. All of these distributions can be uniquely estimated from the observational data, which implies the identifiability of $V^{\pi_t}(s_0)$. Furthermore, $\eta^{\pi_t}$ is identifiable by taking the expectation with respect to the initial state distribution $\nu$.

Here, we provide an example of identifying a linear reward function. Consider the following model: one confounder $U_t$, one auxiliary variable $Z_t$, one state $S_t$, one $d$-dimensional actions are generated as $A_t = \alpha_A U_t + \eta Z_t + \lambda_A S_t$, and one reward generated as $R_t = \alpha_R U_t + \beta_R A_t + \lambda_R S_t$, where $A_t = (A_{t,1}, A_{t,2}, \ldots, A_{t,d})^\top$ and $\alpha_A, \eta, \lambda_A, \beta_R$ are $d$-dimensional vectors of coefficients. In this case, we are interested in obtaining an unbiased estimate of the reward function $R_t$ based on the observed data.

We first estimate $\hat{\eta}$ and $\hat{\lambda_A}$ by regressing $A_t$ on $Z_t$ and $S_t$. Then, we obtain $\hat{\gamma}$ by performing factor analysis on the residuals $A_t - \hat{\eta}Z_t - \hat{\lambda_A}S_t$, where $\hat{\gamma}$ is defined as $(\Sigma_{A_t - \eta Z_t - \lambda_A S_t})^{-1}\alpha_A = (\alpha_A \alpha_A^\top)^{-1}\alpha_A$. This corresponds to step 1, where the confounder distribution is obtained using a linear factor model. We perform a regression of $R_t$ on $Z_t$, $A_t$, and $S_t$, with $(\xi^{Z_t}, \xi^{A_t}, \xi^{S_t})$ represent the coefficients, obtaining to:

$$\xi^{\hat{Z}_t} = -\hat{\gamma}\alpha_R\hat{\eta},$$
$$\xi^{\hat{A}_t} = \hat{\gamma}\alpha_R + \beta_R, \qquad (5)$$
$$\xi^{\hat{S}_t} = \lambda_R - \hat{\gamma}\alpha_R\hat{\lambda_A}.$$

By solving Equation 5, we obtain estimates of the remaining parameters $(\hat{\beta}_R, \hat{\alpha}_R, \hat{\lambda}_R)$. This corresponds to step 2, where the coefficients of $R_t$ are determined by solving the linear equations, thereby identifying the reward function. Finally, we estimate the expected reward under the target policy based on the identified reward function. The state transition function follows a similar process and is not elaborated upon here.

## 5 Estimation

In this section, we demonstrate how to use the Q-function (DM Estimator) to efficiently estimate $\eta^{\pi_t}$. In the context of average cumulative rewards, we define the Q-function as:

$$Q^{\pi_t}(s, a) = \mathbb{E}^{\pi_t}[R_t + V^{\pi_t}(S_{t+1})|S_t = s, A_t = a], \quad (6)$$

where $R_t$ denotes the immediate reward obtained after taking action $A_t$ in state $S_t$, and $V^{\pi_t}(S_{t+1})$ represents the value function at the next state $S_{t+1}$ under the policy $\pi_t$.

Removing the expectation and according to Bellman equation, we obtain that

$$Q^{\pi_t}(s, a) = \sum_{r \in \mathbb{R}} p_r(r|s, a) \cdot r + \\ \sum_{s' \in \mathbb{S}} p_s(s'|s, a) \sum_{a^* \in \mathbb{A}} p_a(a^*|s') Q^{\pi_t}(s', a^*), \quad (7)$$

where $s'$ denotes the next state, $a^*$ the actions taken under policy $\pi_t$, and $p_r(r|s, a)$ and $p_s(s'|s, a)$ can be uniquely identified step by step using historical data and auxiliary variables, as outlined in Theorem 4.1. $Q^{\pi_t}(s', a^*)$ represents the Q-function that follows the target policy in the next state. By aggregating the Q-function over the empirical initial state distribution, we obtain $\eta^{\pi_t} = \mathbb{E}_{S_0 \sim \nu}[Q^{\pi_t}(S_0, A_0)]$.

We now turn to the estimation of $Q^{\pi_t}(s, a)$. Motivated by [31], we employ the Least-Squares Temporal Difference Q-learning (LSTD-Q) method to iteratively solve for the Q-function. Specifically, we begin by using a linear function approximation for the Q-function: $Q^{\pi_t}(s, a; \theta) = \phi(s, a)^\top \theta$, where $\phi(s, a)$ represents the feature vector and $\theta$ is the parameter vector. The temporal difference (TD) error for the Q-function between the state-action pair $(s, a)$ under the behaviour policy and the state-action pair $(s', a^*)$ under the target policy is given by $\delta = r + \phi(s', a^*)^\top \theta - \phi(s, a)^\top \theta$. Using the LSTD-Q method, the following update equation is obtained:

$$\tilde{\mathbf{A}}^{(t+1)} = \tilde{\mathbf{A}}^{(t)} + \phi(s, a)(\phi(s, a) - \phi(s', a^*))^\top, \\ \tilde{b}^{(t+1)} = \tilde{b}^{(t)} + \phi(s, a)r, \quad (8)$$

where $\mathbf{A}$ denotes the sum of the covariance matrices for the state-action pairs, and $b$ represents the accumulation of each state-action pair, weighted by the corresponding immediate reward.

Finally, we update the parameter $\theta$ by solving the equation $\theta = \mathbf{A}^{-1}b$. A more detailed derivation of the LSTD-Q method can be found in [31].

## 6 Experiments

In this section, we evaluate the performance of the proposed estimator through a simulation experiment and an experiment involving autistic children based on a real OPE application.

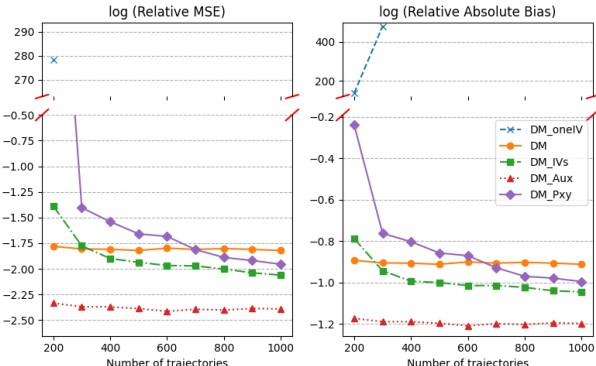

**Figure 3: Logarithmic relative MSE in the left half and logarithmic relative absolute bias in the right half of the figure, with sample size on the x-axis.**

### 6.1 Simulations

We compare the proposed estimator with several baseline methods using synthetic data.

**Data generating process.** We begin by describing the detailed setup for the simulation. The observed data consists of $N = 1000$ trajectories, each with $T = 50$ time steps. The unobserved confounders $\{U_t\}_{t=1}^T$ are independently and identically distributed (i.i.d.), sampled from a standard normal distribution $\mathcal{N}(0, 1)$. The confounder proxy is generated as $W_t = 2U_t$. At each time step $t$, six actions, $A_t = (A_{t,1}, A_{t,2}, \ldots, A_{t,6})$, are assigned according to the behaviour policy, which satisfies $A_{t,i} = Z_{t,i} + S_t + U_t$ for $i \in 1, 2, \ldots, 6$. Here, $Z_t = (Z_{t,1}, Z_{t,2}, \ldots, Z_{t,6})$ denotes six instrumental variables, drawn from a 6-dimensional multivariate normal distribution $Z_t \sim \mathcal{N}(0, \mathbf{I}_6)$, corresponding to each of the six actions. One of these variables is taken as the auxiliary variable. The reward function and state transition function are defined as: $R_t = \sum_{i=1}^6 A_{t,i} + S_t + 2.5U_t$, and $S_{t+1} = (\sum_{i=1}^6 A_{t,i} + S_t)/10 + 5U_t$. The initial state $S_0$ is also sampled from $\mathcal{N}(0, 1)$.

**Compared methods.** We consider four baseline estimators. The first is a direct method (DM) that ignores the presence of confounders, where the Q-function is used directly to estimate the target policy. The second approach combines the Q-function with instrumental variables (DM_IVs) [42]. Based on the completeness assumption of instrumental variables [45], this method requires as many instrumental variables as there are actions, specifically, $Z_t = (Z_{t,1}, Z_{t,2}, \ldots, Z_{t,6})$. To ensure fairness, the third approach (DM_oneIV) uses only one instrumental variable from $Z_t$ in the Q-function, aligning with the number of auxiliary variables required by our proposed estimator. The fourth approach combines Q-functions with confounder proxies (DM_Pxy) [40, 60]. We use one of $Z_t$ and $W_t$ as treatment- and outcome-inducing confounder proxies, respectively. Given the target policy, which takes six actions, with all values set to 1 at each time step, i.e., $A_{t\,t=1}^T = (1, 1, 1, 1, 1, 1)_{t=1}^T$, we use the above estimators to evaluate it.

**Results.** We use logarithmic relative MSE (logMSE) and logarithmic relative absolute bias (logBias) as evaluation metrics, with the ground truth being $\eta^{\pi_t}$ obtained by following the target policy in the unconfounded MDP. Each experiment was repeated 100 times

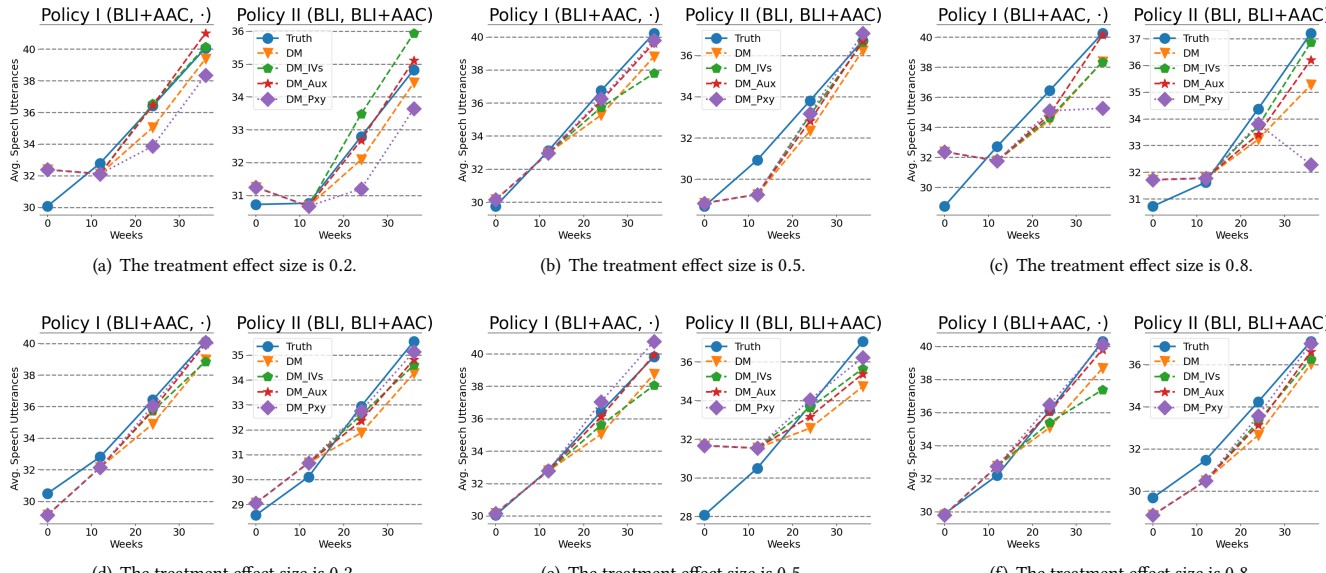

(a) The treatment effect size is 0.2.

(b) The treatment effect size is 0.5.

(c) The treatment effect size is 0.8.

(d) The treatment effect size is 0.2.

(e) The treatment effect size is 0.5.

(f) The treatment effect size is 0.8.

**Figure 4: Evaluation results of autistic children simulation with different treatment effect sizes under different sample sizes (The strengths of the confounding $\gamma = 1$). The x-axis represents the number of weeks, indicating the progression of time during the treatment or intervention period. The y-axis denotes the mean count of verbal expressions made by children with autism throughout the course of the treatment. The top row represents a sample size of 100, and the bottom row represents a sample size of 1000.**

across different numbers of trajectories. Figure 3 summarizes the bias of these estimators. Our proposed estimator (DM_Aux) performs well, achieving the smallest logMSE and logBias compared to the baseline methods. In contrast, DM_oneIV produces outlier values due to the incorrect number of instrumental variables, highlighting potential issues with the estimator's robustness or stability when there are insufficient instrumental variables. Our estimator effectively addresses this issue.

Additionally, we find that the traditional IV-based estimator (DM_IVs) and the confounder proxy-based estimator (DM_Pxy) experience increases in logMSE and logBias as the number of trajectories decreases, which aligns with the asymptotic properties of IV and proxy methods [17, 58, 60]. This demonstrates the limitations of IV- or proxy-based estimators with small sample sizes, whereas our proposed method performs better in such cases. Furthermore, the DM estimator suffers from significant bias in its estimates, as it does not account for the presence of unobserved confounders.

## 6.2 Autism example

In this section, we apply our method to a treatment recommendation example: communication interventions for minimally verbal children with autism. Minimally verbal children makeup 25-30% of those with autism and often have a poor prognosis in terms of social functioning. Using a simulator for autistic children developed by Lu et al. [37], which models data from a Sequential Multiple Assignment Randomized Trial (SMART) [24], we evaluated the treatment effects (measured by the number of socially communicative utterances) under different treatment regimes (target policies).

**Overview.** In the autism SMART trial, there are two therapeutic interventions (multiple actions): a therapist's behavioural language intervention (BLI) and a device for augmented/alternative communication (AAC). We consider treatment provider preferences, such as the conversation content of BLI and the device assignment of AAC determined by clinicians, as instrumental or auxiliary variables [13, 44]. Actions are taken at weeks 12, 24, and 36 ($T = 2, 3, 4$), and the number of speech utterances is measured in weeks 24 and 36 ($T = 3, 4$). The average number of speech utterances among autistic children serves as the reward or outcome. In the original study [37], two treatment policies were evaluated (listed in Table 1 of the original article). However, there may be slight confounding due to unrecorded patient information, such as the foundational cognitive abilities of the patients.

**Real data collection.** Following Lu et al. [37], the data generation process in the autistic children experiment is based on a sample of 300 individuals from Kasari et al. [24]. Each patient is characterized by six covariates: age, gender, and indicators for African American, Caucasian, Hispanic, and Asian. To obtain a sample size of $N$, we sample with replacement from this set.

**Actions and target policy.** In the autism SMART trial, two actions are available at weeks 24 and 36 ($T = 3, 4$): $A_1 \in \{-1, 1\}$ and $A_2 \in \{-1, 1\}$. Here, $a_1 = 1$ denotes BLI, while $a_1 = -1$ denotes BLI+AAC. Similarly, $a_2 = 1$ represents assigning intensified BLI, and $a_2 = -1$ represents assigning BLI+AAC. Although $A_1$ and $A_2$ are two-stage treatments in the original study, we treat these as multiple actions assigned at week 24 and week 36, based on the outcome equation 9. Additionally, we focus on children with slow

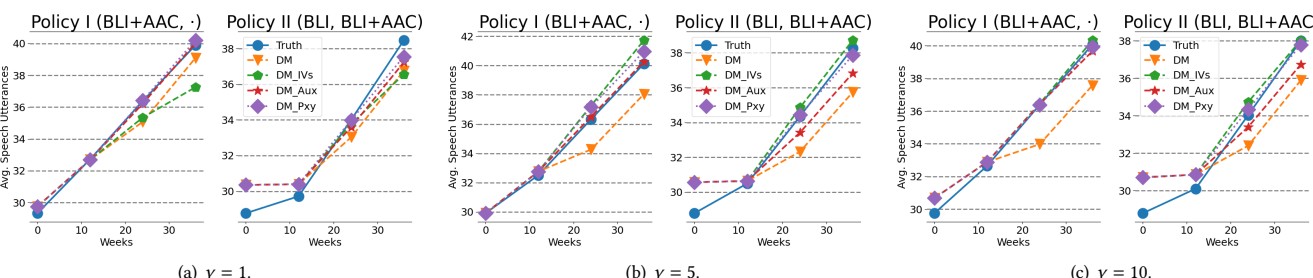

(a) $\gamma = 1$.

(b) $\gamma = 5$.

(c) $\gamma = 10$.

**Figure 5: Evaluation results of autistic children simulation with different strengths of the confounding under sample size of 1000 (The treatment effect sizes are all set to** $0.95$**).**

responses to ensure multiple treatments are administered, i.e., $R$ is always equal to 0 in the outcome equation 9. Further details can be found in the original work [37].

There are two different target policies to evaluate. Policy I: using AAC from the beginning (BLI+AAC, ·). Policy II: deferring the use of AAC (BLI, BLI+AAC).

**Confounding.** The original simulator did not include unobserved confounders. Here, we describe how confounding is introduced in this setting.

Lu et al. [37] gives the effect of two treatments on the reward outcome $Y_{24}$ and $Y_{36}$ as follows:

$$
\begin{aligned}
Y_{24} = {} & \eta_{21}^T X + \eta_{22} Y_0 + \eta_{23}^T A_1 \\
& + \eta_{24} Y_{12} + \beta_{21}(1-R)(A_1+1)A_2 + \epsilon_2 \\
Y_{36} = {} & \eta_{31}^T X + \eta_{32} Y_0 + \eta_{33}^T A_1 \\
& + \eta_{34} Y_{12} + \beta_{31}(1-R)(A_1+1)A_2 + \epsilon_3
\end{aligned}
\tag{9}
$$

where $\eta_{23}$, $\beta_{21}$ and $\eta_{33}$, $\beta_{31}$ can be regarded as effect size of two treatments on $Y_{24}$ and $Y_{36}$, respectively.

In the original setting, the authors generated four treatment effects with different numerical sizes (Figure 7 of the original text). We give evaluation results for different treatment effect sizes (excluding effect size of 0) in the presence of unobserved confounders, as shown in Figure 4. The definitions and specific values of remaining parameters in this simulation are reported by [37].

We introduce confounding by adding $U$ that follows the discrete uniform distribution, to the outcome model, i.e. Equation 9, respectively. This is because some of the baseline methods (IVs) require additional assumptions, such as an additive outcome model, which does not allow treatment and confounder to have an interaction, i.e., $E(Y|u,x) = m(x) + u$ [45]. Our method is not subject to this restriction [41]. More precisely, we randomly assign $U$ to either $\gamma$ or $-\gamma$ for each individual, where $\gamma$ controls the strength of the confounding effect. We also show the evaluation results for two target policies under different strengths of the confounding, as shown in Figure 5.

**Behaviour Policy and Auxiliary Variable** In the original work [37], two actions $A_1$, $A_2$ are taken according to a random policy, i.e. $P(A_1 = -1) = P(A_1 = 1) = 0.5$ and $P(A_2 = -1) = P(A_2 = 1) = 0.5$. In our experiments, we specify that two actions are taken according to the behaviour policy $A_1 \sim \pi_b(Z_1, U) = Z_1 + U + \epsilon$ and $A_2 \sim \pi_b(Z_2, U) = Z_2 + U + \epsilon$, where $Z_1$ and $Z_2$ denote instrumental

variables or auxiliary variables. Here, the practical significance of $Z_1$ is the content of the conversation prescribed by the clinician, and $Z_2$ is the assignment of devices decided by the clinician.

**Results** The results of the estimation for the two target policies are reported in Figure 4-5. Each set of experiments was repeated 100 times. Compared to other estimators, our estimator yields estimates that more closely align with the ground truth curve under various parameter settings, demonstrating it can effectively handle the bias introduced by unobserved confounders. Moreover, in the control group with a smaller sample size, our proposed estimator delivers more accurate estimates, while other baseline methods yield results even worse than DM, which does not account for confounders. This highlights the advantage of our approach, particularly in settings with limited sample sizes. Although our method may slightly underperform compared to some idealized approaches in a few specific parameter settings, it requires significantly fewer auxiliary variables. This makes our estimator more feasible to implement in real-world scenarios, highlighting its practical applicability and potential for broader adoption.

## 7 Conclusion

In this paper, we present a systematic approach to evaluate off-policy using auxiliary variables in the presence of unobserved confounders in multi-action scenarios. Our approach overcomes the limitations of traditional auxiliary variable methods for multi-action scenarios by requiring only a single auxiliary variable, relaxing the need for as many auxiliary variables as the actions. The experimental results in simulation and examples of autistic children demonstrate the effectiveness of our proposed approach. To the best of our knowledge, this is the first work to address the presence of an unobserved confounder in offline multi-action policy evaluation.

The estimator of the direct method relies on the correct specification of the Q-function. If the Q-function is misspecified, the results of the evaluation may be affected. This leads to several potential future works that could build on this paper: One possibility is to extend the direct method to the doubly robust technique in OPE, drawing on the strengths of two or more estimators to overcome the problem of misspecified Q-functions. Another option is to make use of deep neural networks, such as deep Q-learning, which can be an effective way to avoid specifying Q-function in the face of unknown, complex data generation processes.

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

# A Appendix

This is the Appendix for "Off-policy Evaluation for Multiple Actions in the Presence of Unobserved Confounders".

## A.1 Proof of Theorem 1

Theorem 4.1 states the identifiability of the value function, i.e. $V^{\pi_t}(s_0)$ can be unbiasedly estimated from the observed data even in the presence of unobserved confounders.

As can be seen from Equation 2, the solution of the value function is an iterative process, which suffices to identify the immediate reward $E^{\pi_t}[R_t|S_0 = s_0]$ at each time step $t$. Therefore, according to Assumption 1, Equation 2 can be further decomposed as

$$
\begin{aligned}
E^{\pi_t}[R_t|S_0 = s_0] = \sum_{s_0 \in \mathcal{S}} \sum_{s_1 \in \mathcal{S}} \cdots \sum_{s_t \in \mathcal{S}} R_t \cdot \\
\mathbb{P}(R_t|do(A_t = \pi(s_t)), S_t = s_t, U_t = u_t) \cdot \\
\mathbb{P}(S_t|do(A_{t-1} = \pi(s_{t-1})), S_{t-1} = s_{t-1}, U_{t-1} = \\
u_{t-1}) \cdots \mathbb{P}(S_1|do(A_0 = \pi(s_0)), S_0 = s_0, U_0 = u_0)
\end{aligned}
\tag{10}
$$

where $do(A_j = \pi(s_j)) = do(A_{j,1} = \pi_t(s_j), \dots, A_{j,d} = \pi_t(s_j)), 0 \le j \le t$ denotes the actions taken under the target policy $\pi_t$ for any $t \in [T]$.

According to 10, the identification procedure of $V^{\pi_t}(s_0)$ can be conducted stage-by-stage. In the following, we will identify each term on the right-hand side of Equation 10 in three steps.

**Step 1.** Identifiability of $\mathbb{P}(S_j = s_j|do(A_{j-1} = \pi(s_{j-1})), S_{j-1} = s_{j-1}, U_{j-1} = u_{j-1}), \forall 1 \le j \le t$.

According to the back-door adjustment [48], the potential state distribution is identified by

$$
\begin{aligned}
&\mathbb{P}(S_j = s_j|do(A_{j-1} = \pi(s_{j-1})), S_{j-1} = s_{j-1}, U_{j-1} = u_{j-1}) \\
&= \sum_{s_{j-1}} \sum_{u_{j-1}} p(s_j|s_{j-1}, a_{j-1}, u_{j-1}) p(s_{j-1}) p(u_{j-1}).
\end{aligned}
\tag{11}
$$

Here, the state transition distribution $p(s_j|s_{j-1}, a_{j-1}, u_{j-1})$ and the product of the probability distributions of $p(s_{j-1})$ and $p(u_{j-1})$ need to be identified separately.

We first identify the state transition distribution $p(s_j|s_{j-1}, a_{j-1}, u_{j-1})$ by equation

$$
\begin{aligned}
&p(s_j|s_{j-1}, a_{j-1}, z_{j-1}) \\
&= \sum_{u_{j-1}} p(s_j|s_{j-1}, a_{j-1}, u_{j-1}) p(u_{j-1}|s_{j-1}, a_{j-1}, z_{j-1}).
\end{aligned}
\tag{12}
$$

(See Equations 13 - 16 for derivation of Equation 12), where $p(s_j|s_{j-1}, a_{j-1}, z_{j-1})$ is the distribution function for the observable random vector $(s_j, s_{j-1}, a_{j-1}, z_{j-1})$, which can be estimated parametrically or non-parametrically using standard density estimation techniques. Examples include parameter estimation to describe observed variables whose distribution form is known, or density estimation based directly on observed variables without assuming the distribution form; $p(u_{j-1}|s_{j-1}, a_{j-1}, z_{j-1})$ is the distribution of $u_{j-1}$ given $s_{j-1}, a_{j-1}, z_{j-1}$. This step needs Assumption 2 ii. To estimate this distribution, we need to correctly specify a state-actions-confounder model that meets Assumption 2 (ii), such as a factor model. Under a standard factor model, estimation of $p(u_{j-1}|s_{j-1}, a_{j-1}, z_{j-1})$ is well established. There is extensive literature on the estimation technique available here [1] [66]. By solving

Equation 12, we obtain the unique solution for the state transition function $p(s_j|s_{j-1}, a_{j-1}, u_{j-1})$ (See Remark 4. for the reason). Notably, this process does not require observing the variable $u_{j-1}$; the identification can be achieved solely based on other observable variables.

The challenging part of this step is the right-hand side of Equation 12, solving the integral equation, which usually has no closed-form solution. This kind of equation is the form of Fredholm integral equations of the first kind [28] and is known to be ill-posed due to the noncontinuity of the solution. The numerical solution of such equations is an active field of research in mathematics and statistics, and goes beyond the scope of this discussion. However, we note that [8] and [41] provide a consistent estimator of certain parametric models under mild conditions, obviating the need to solve integral equation (See these two works for more details and proof).

The following part shows the derivation of Equation 12. Consider the joint distribution $p(s_j, s_{j-1}, a_{j-1}, z_{j-1}, u_{j-1})$, according to Bayes' law, we decompose this distribution recursively as

$$
\begin{aligned}
&p(s_j, s_{j-1}, a_{j-1}, z_{j-1}, u_{j-1}) \\
&= p(s_j|s_{j-1}, a_{j-1}, z_{j-1}, u_{j-1})p(s_{j-1}, a_{j-1}, z_{j-1}, u_{j-1}) \\
&= p(s_j|s_{j-1}, a_{j-1}, z_{j-1}, u_{j-1})p(u_{j-1}|s_{j-1}, a_{j-1}, z_{j-1}) \cdot \\
&\quad p(s_{j-1}, a_{j-1}, z_{j-1}).
\end{aligned} \tag{13}
$$

Moving $p(s_{j-1}, a_{j-1}, z_{j-1})$ to the left-hand side, the joint distribution of $s_j$ and $u_{j-1}$ can be derived as

$$
\begin{aligned}
&p(s_j, u_{j-1}|s_{j-1}, a_{j-1}, z_{j-1}) \\
&= p(s_j|s_{j-1}, a_{j-1}, z_{j-1}, u_{j-1})p(u_{j-1}|s_{j-1}, a_{j-1}, z_{j-1}).
\end{aligned} \tag{14}
$$

According Assumption 2 i, the equation can be written as

$$
\begin{aligned}
&p(s_j, u_{j-1}|s_{j-1}, a_{j-1}, z_{j-1}) \\
&= p(s_j|s_{j-1}, a_{j-1}, u_{j-1})p(u_{j-1}|s_{j-1}, a_{j-1}, z_{j-1}).
\end{aligned} \tag{15}
$$

To obtain $p(s_j|s_{j-1}, a_{j-1}, z_{j-1})$, we perform marginalization over $u_{j-1}$ with respect to Equation 15:

$$
\begin{aligned}
p(s_j|s_{j-1}, a_{j-1}, z_{j-1}) &= \sum_{u_{j-1}} p(s_j, u_{j-1}|s_{j-1}, a_{j-1}, z_{j-1}) \\
&= \sum_{u_{j-1}} p(s_j|s_{j-1}, a_{j-1}, u_{j-1})p(u_{j-1}|s_{j-1}, a_{j-1}, z_{j-1}).
\end{aligned} \tag{16}
$$

Next, we identify the product of the probability distributions of $p(s_{j-1})$ and $p(u_{j-1})$ in Equation 11. We again use Bayes' Law for the joint distribution $p(s_{j-1}, a_{j-1}, z_{j-1}, u_{j-1})$ to obtain Equation 17, and then marginalize over $a_{j-1}$ and $z_{j-1}$ for $p(s_{j-1}, a_{j-1}, z_{j-1}, u_{j-1})$ to obtain Equation 18:

$$
p(s_{j-1}, a_{j-1}, u_{j-1}, z_{j-1}) = p(s_{j-1}, a_{j-1}, u_{j-1}|z_{j-1})p(z_{j-1}), \tag{17}
$$

$$
p(s_{j-1}, u_{j-1}) = \sum_{z_{j-1}} \sum_{a_{j-1}} p(s_{j-1}, a_{j-1}, u_{j-1}, z_{j-1}). \tag{18}
$$

Solve them simultaneously to obtain:

$$
p(s_{j-1}, u_{j-1}) = \sum_{z_{j-1}} \sum_{a_{j-1}} p(s_{j-1}, a_{j-1}, u_{j-1}|z_{j-1})p(z_{j-1}). \tag{19}
$$

Here, the nodes of $S_{j-1}$ and $U_{j-1}$ satisfy the *Collider* structure [50] (See Figure 2) and are independent when the variables $A_{j-1}$, $R_{j-1}$, and $S_j$ are not given [50]. Thus, the joint distribution of $s_{j-1}$

and $u_{j-1}$ can be written as a product of their respective marginal distributions:

$$
p(s_{j-1})p(u_{j-1}) = \sum_{z_{j-1}} \sum_{a_{j-1}} p(s_{j-1}, a_{j-1}, u_{j-1}|z_{j-1})p(z_{j-1}). \tag{20}
$$

Analogously to the identification of $p(u_{j-1}|s_{j-1}, a_{j-1}, z_{j-1})$ from Equation 12, Assumption 2 ii and iii restrict the state-actions-confounder distribution $p(s_{j-1}, a_{j-1}, u_{j-1}|z_{j-1})$, which allows that $p(s_{j-1}, a_{j-1}, u_{j-1}|z_{j-1})$ to be determined by a factor model.

Substituting Equation 20 into Equation 11, the identification result of $\mathbb{P}(S_j = s_j|do(A_{j-1} = \pi(s_{j-1})), S_{j-1} = s_{j-1}, U_{j-1} = u_{j-1}), \forall 1 \le j \le t$ is given by

$$
\begin{aligned}
&\mathbb{P}(S_j = s_j|do(A_{j-1} = \pi(s_{j-1})), S_{j-1} = s_{j-1}, U_{j-1} = u_{j-1}) \\
&= \sum_{z_{j-1}, s_{j-1}, a_{j-1}, u_{j-1}} p(s_j|s_{j-1}, a_{j-1}, u_{j-1})p(s_{j-1}, a_{j-1}, u_{j-1}|z_{j-1}) \cdot \\
&\quad p(z_{j-1}).
\end{aligned} \tag{21}
$$

Equation 21 can be considered as the identification process in a single time step, in which all probability functions can be identified consistently based on the observed data.

**Step 2.** Identifiability of $\mathbb{P}(R_t = r_t|do(A_t = \pi(s_t)), S_t = s_t, U_t = u_t)$.

As shown in the causal graph in Figure 2, $R_t$ and $S_{t+1}$ have the same causal hierarchy. Thus, The identification of $\mathbb{P}(R_t = r_t|do(A_t = \pi(s_t)), S_t = s_t, U_t = u_t)$ can be easily written as

$$
\begin{aligned}
&\mathbb{P}(R_t = r_t|do(A_t = \pi(s_t)), S_t = s_t, U_t = u_t) \\
&= \sum_{z_t, s_t, a_t, u_t} p(r_t|s_t, a_t, u_t)p(s_t, a_t, u_t|z_t)p(z_t).
\end{aligned} \tag{22}
$$

**Step 3.** Repeating **Step 1.** from $j = 0$ to $j = t$, we can obtain the expectation of potential reward at step $t$

$$
\begin{aligned}
E^{\pi_t}[R_t|S_0 = s_0] &= \sum_{\{z_j, a_j, u_j, r_j, s_{j+1}\}_{j=0}^t} r_t \cdot \\
&\{\prod_{j=0}^t p_{s,r}(s_{j+1}, r_j|s_j, a_j, u_j) \cdot p_{s,a,u}(s_j, a_j, u_j|z_j) \cdot p_z(z_j)\}.
\end{aligned} \tag{23}
$$

Therefore, the value function $V^{\pi_t}(s_0)$ can be written as

$$
\begin{aligned}
V^{\pi_t}(s_0) &= \frac{1}{T} \sum_{t=0}^T E^\pi[R_t|S_0 = s_0] = \frac{1}{T} \sum_{t=0}^T \sum_{\{z_j, a_j, u_j, r_j, s_{j+1}\}_{j=0}^t} r_t \cdot \\
&\{\prod_{j=0}^t p_{s,r}(s_{j+1}, r_j|s_j, a_j, u_j) \cdot p_{s,a,u}(s_j, a_j, u_j|z_j) \cdot p_z(z_j)\}.
\end{aligned} \tag{24}
$$

Furthermore, the identification result of $\eta^{\pi_t}$ can be obtained by taking the expectation of $V^{\pi_t}(s_0)$ on the initial state distribution $\nu(s_0)$, which is given by

$$
\begin{aligned}
\eta^{\pi_t} &= \sum_{s_0} [\frac{1}{T} \sum_{t=0}^T \sum_{\{z_j, a_j, u_j, r_j, s_{j+1}\}_{j=0}^t} r_t \cdot \\
&\{\prod_{j=0}^t p_{s,r}(s_{j+1}, r_j|s_j, a_j, u_j) \cdot p_{s,a,u}(s_j, a_j, u_j|z_j) \cdot p_z(z_j)\}]\nu(s_0).
\end{aligned} \tag{25}
$$

The proof is thus completed.

We say that $p(s_j|s_{j-1}, a_{j-1}, u_{j-1})$ is uniquely determined from Equation 12. This is because $p(u_{j-1}|s_{j-1}, a_{j-1}, z_{j-1})$ is complete in $z_{j-1}$ under Assumption 2 (iii). If there is more than one candidate solution for Equation 12, e.g., $p_1^*(s_j|s_{j-1}, a_{j-1}, u_{j-1})$ and $p_2^*(s_j|s_{j-1}, a_{j-1}, u_{j-1})$, such that $\sum_{u_{j-1}}\{p_1^*(s_j|s_{j-1}, a_{j-1}, u_{j-1}) - p_2^*(s_j|s_{j-1}, a_{j-1}, u_{j-1})\}p(u_{j-1}|s_{j-1}, a_{j-1}, z_{j-1})$ is not equal to 0, this violates the completeness of $p(u_{j-1}|s_{j-1}, a_{j-1}, z_{j-1})$ in $z_{j-1}$. Therefore, $p(s_j|s_{j-1}, a_{j-1}, u_{j-1})$ is uniquely determined from Equation 12, and $\mathbb{P}(S_j = s_j|do(A_{j-1} = \pi(s_{j-1})), S_{j-1} = s_{j-1}, U_{j-1} = u_{j-1})$ is identified by plugging it into 21.

## A.2 Discounted Cumulative Reward

We extend our proposal to the setting of discounted cumulative rewards in this section.

Given a discount factor $0 \le \gamma < 1$, the value function $V^{\pi_t}(s_0)$ is defined as the expected sum of rewards, each weighted by a discount factor, starting from an initial state under a target policy:

$$V^{\pi_t}(s_0) = \sum_{t=0}^{T} \gamma^t \mathbb{E}^{\pi_t}[R_t|S_0 = s_0]. \tag{26}$$

Referring to the identification process A.1 under the average reward setting, we can easily obtain the identification results under the discounted reward setting:

$$V^{\pi_t}(s_0)$$
$$= \sum_{t=0}^{T} \sum_{\tau_t} \gamma^t r_t \{\prod_{j=0}^{t} p_{s,r}(s_{j+1}, r_j|s_j, a_j, u_j)p_{s,a,u}(s_j, a_j, u_j|z_j)p_z(z_j)\} \tag{27}$$

We extend the direct method to the policy value estimator under discounted reward setting, the Q-function is defined as:

$$Q^{\pi_t}(s, a) = \mathbb{E}^{\pi_t}[R_t + \gamma V^{\pi_t}(S_{t+1})|S_t = s, A_t = a], \tag{28}$$

We then expand it according to the Bellman equation:

$$Q^{\pi_t}(s, a) = \sum_{r \in \mathbb{R}} p_r(r|s, a) \cdot r +$$
$$\gamma \sum_{s' \in \mathbb{S}} p_s(s'|s, a) \sum_{a^* \in \mathbb{A}} p_a(a^*|s')Q^{\pi_t}(s', a^*), \tag{29}$$

where the identification of $p_r(r|s, a)$ and $p_s(s'|s, a)$ are consistent with those in the main paper.

We next discuss the estimating procedures of $Q^{\pi_t}(s, a)$ using LSTD-Q method [31]. The TD error of the Q-function in the discounted reward setting is given by $\delta = r + \gamma\phi(s', a^*)^\top\theta - \phi(s, a)^\top\theta$. The update equation can be rewritten as

$$\tilde{\mathbf{A}}^{(t+1)} = \tilde{\mathbf{A}}^{(t)} + \phi(s, a)(\phi(s, a) - \gamma\phi(s', a^*))^\top,$$
$$\tilde{b}^{(t+1)} = \tilde{b}^{(t)} + \phi(s, a)r. \tag{30}$$

Received 20 February 2007; revised 12 March 2009; accepted 5 June 2009

