# OpenReview forum: "Off-policy Evaluation for Multiple Actions in the Presence of Unobserved Confounders"
_ACM.org/TheWebConf/2025/Conference — WWW 2025 Oral_

### Official Review · Reviewer_GBwh · 2024-11-25

**Novelty:** 4
**Technical Quality:** 5

**Review:**

Summary:
This paper addresses the significant challenge of off-policy evaluation (OPE) in reinforcement learning, specifically in scenarios with multi-action settings and the presence of unobserved confounders. The authors introduce a novel method that employs a single auxiliary variable to mitigate biases in estimation caused by these unobserved factors. This is a departure from existing methods that are less effective in multi-action contexts, often requiring a number of auxiliary variables proportional to the number of actions available. The theoretical analysis provided in the paper supports the claim that this new method delivers unbiased estimations of the target policy value. Furthermore, empirical evaluations in both simulated experiments and a real-world treatment recommendation application demonstrate the proposed method’s superior performance compared to current baseline methods. The study’s contributions include the development of this auxiliary variable-based method for unbiased OPE in multi-action settings with unobserved confounders, a thorough theoretical validation of its unbiased estimation capability, and empirical evidence of its effectiveness and reliability.

**Questions:**

Strengths:

1.The paper addresses off-policy evaluation in the multi-action setup with unobserved confounders, which is pioneering.
2.The manuscript is well-organized. The author provides a very detailed introduction to the research background. There is a thorough theoretical analysis of the proposed method.
3.The effectiveness of the proposed method has been demonstrated in both simulation experiments and real-world application scenarios.

Weaknesses:

1. The article could consider adding an overview diagram for the proposed method to visually present the approach.
2. Since the article did not conduct comparative experiments on public datasets, the authors should provide more explanations regarding the rationale behind their designed simulation experiments. I believe the authors should clarify the details of the simulation more clearly, such as the setting of various hyper-parameters and the design of evaluation metrics. Some necessary relevant references may also need to be added.
3. The authors could consider adding specific cases from the autism treatment recommendation experiment to demonstrate the model's effectiveness.

**Reviewer Confidence:**

2: The reviewer is willing to defend the evaluation, but it is likely that the reviewer did not understand parts of the paper

**Scope:**

3: The work is somewhat relevant to the Web and to the track, and is of narrow interest to a sub-community

---

### Official Review · Reviewer_4UAG · 2024-11-29

**Novelty:** 5
**Technical Quality:** 4

**Review:**

Summary:
This paper presents a novel method for off-policy evaluation (OPE) in multi-action reinforcement learning, addressing the challenges of unobserved confounders. While existing OPE methods primarily focus on single-action settings, this work proposes an auxiliary variable-aided approach tailored for multi-action scenarios. The method claims to achieve unbiased policy value estimation using a single auxiliary variable, relaxing restrictive assumptions found in prior work. Theoretical analysis and empirical evaluations, including synthetic and real-world treatment recommendation experiments, demonstrate the effectiveness of the proposed approach.

Pros:
- The paper is well-written and the relationship and improvement relative to previous work are clearly explained.
- The paper addresses a significant gap in OPE by tackling multi-action scenarios with unobserved confounders using a single auxiliary variable, an area largely overlooked in existing literature.
- Empirical evaluations demonstrate the effectiveness of the proposed approach.

Cons:
- Limited focus on Web-related scenarios, which reduces its relevance for this venue.

**Questions:**

- While treatment recommendation is important, it appears to have limited relevance to Web-related scenarios. Beyond the diabetes and autism examples, could the authors add examples of other real-world multi-action tasks more closely related to Web applications?
- Could the authors elaborate on how auxiliary variables are selected in practice? It seems remaining unclear how these variables are identified or validated in real-world scenarios.
- The experimental results lack confidence intervals or error bars.  Could the authors include it to clarify the results?

**Reviewer Confidence:**

1: The reviewer's evaluation is an educated guess

**Scope:**

1: The work is irrelevant to the Web

---

### Official Review · Reviewer_kUCz · 2024-12-01

**Novelty:** 4
**Technical Quality:** 4

**Review:**

This paper extends auxiliary variable aided method to identify effects of multiple treatments in the presence of
unmeasured confounding for the off-policy evaluation. This is a significant expansion of traditional causal inference methods to RL.

Presentation is somewhat clear with solid mathematical proofs.
The section studying real-world data shows that this method might be practically relevant.

Weaknesses:
Paper assumes static adjustment for confounders at each time step using auxiliary variables but does not model evolution of confounders . If confounders evolve dynamically, static adjustment may leave residual confounding unaddressed, leading to potentially biased estimates of state transitions, rewards and policy value as well as error accumulation.

**Questions:**

The temporal difference (TD) error update in LSTD-Q uses linear approximations. Linear approximations might not capture complex dynamics in multi-action scenarios.  How does the paper justify its choice of linearity?

In practical applications, some actions might be  unobserved for specific states or contexts. How this approach would deal with such situations?

Factor models are used to estimate the latent confounder. These models can be highly sensitive to the number of factors and initialization. How robust  is estimated policy value if the factor model is misspecified?

**Reviewer Confidence:**

2: The reviewer is willing to defend the evaluation, but it is likely that the reviewer did not understand parts of the paper

**Scope:**

3: The work is somewhat relevant to the Web and to the track, and is of narrow interest to a sub-community

---

### Official Review · Reviewer_tgc1 · 2024-12-02

**Novelty:** 5
**Technical Quality:** 4

**Review:**

This paper addresses the problem of Off-Policy Evaluation (OPE) in multi-action decision-making scenarios, specifically when unobserved confounders are present.
The authors propose a novel approach leveraging auxiliary variables and provide a robust framework for unbiased policy evaluation.
Unlike traditional methods, DM_Aux requires only a single auxiliary variable for multi-action scenarios.
The paper is grounded in the counterfactual or potential outcomes framework, extended to decision-making in Markov Decision Processes (MDPs).

**Questions:**

1. The Exclusion Restriction in asumption seems to be very strong. In practice, identifying valid auxiliary variables can be challenging. If an auxiliary variable directly influences the reward (violating exclusion restriction), the method may produce biased estimates.

2. As mentioned in line 616, The Q-function is approximated with a linear function, which might limit the scablity of the proposed estimation framework.

3. The experiments are conducted with a simulation and another example of autism, which are more like toy examples with ralative small data size. The lack of large scale real-world data might restrict the application of the proposal.

**Reviewer Confidence:**

3: The reviewer is confident but not certain that the evaluation is correct

**Scope:**

4: The work is relevant to the Web and to the track, and is of broad interest to the community

---

### Official Review · Reviewer_WBNo · 2024-12-02

**Novelty:** 5
**Technical Quality:** 5

**Review:**

Summary: The authors add to the literature on off policy evaluation in the presence of unobserved confounders by providing a method which can be applied in cases where the decision process of interest includes multiple actions as part of each decision.

*Quality/Originality/Significance*

---
Pros:
- From my reading, this is a solid paper. It uses causal inference techniques to solve a problem in reinforcement learning. I think this is valuable as not all of us in RL research have a background in causal inference (myself included; to be transparent, this limits my ability to review this work), but certainly all of us in RL research that want to work on any kind of practical problem will need to reckon with problems related to confoundedness.
- The authors did a good job of establishing the novelty and applicability of their work via by citing and discussing prior works.
- I appreciate that the authors used examples in healthcare to motivate the problem and as part of the experiments. I like when a paper is technical *and* shows how its technical contribution can be employed in practice.
- Fig 3 shows that the proposed method clearly outperforms the baselines, at least with respect to the chosen metrics

Cons:
- Authors may want to give a stronger justification of why OPE is important and evidence that it actually is a "key topic in reinforcement learning"
- Authors may want to compare OPE with offline RL in the related work section
- I would have appreciated justification of why log relative MSE and log bias are good choices of metrics
- It is not immediately clear to me from Figs 4/5 that the proposed method systematically outperforms the baselines

*Clarity*

---

Pros:
- Followed the familiar format of a CS conference paper. In other words, the organization of the paper made it easy to follow.
- I appreciated how comprehensive the intro and literature review was

Cons:
- Some text is difficult to read (e.g. in Figure 5)
- It is difficult to tell the difference in the lines plotted in figures 4,5
- I wanted more explanation of what the "simulator" was in the Autism example (line 749)
- Authors may want to define/characterize auxiliary variables early on in the paper, since it is a key ingredient of the paper
- I think there may be a grammar issue on lines 400-401

**Questions:**

1. Can you give a more thorough explanation of how this paper differs from the work of Miao et al. (reference 41)?
2. What is your justification for using log relative MSE and log bias?
3. When you refer to repeating experiments 100 times, what is different across the 100 runs? What is the same?
4. What were you hoping readers would take away from looking at figures 4 and 5?

**Reviewer Confidence:**

2: The reviewer is willing to defend the evaluation, but it is likely that the reviewer did not understand parts of the paper

**Scope:**

3: The work is somewhat relevant to the Web and to the track, and is of narrow interest to a sub-community